# OpenReview forum: "How Susceptible are LLMs to Influence in Prompts?"
_colmweb.org/COLM/2024/Conference — COLM_

### Official Review · Reviewer_LLEW · 2024-05-06

**Rating:** 7
**Confidence:** 3
**Ethics Flag:** 1

**Summary:**

This paper investigates the issue of the influence of augmented prompts on the accuracy of LLMs in a question answering setting, with the augmented input being provided by another advocate model (in this case, LLama-2). The authors test three different LLMs (Mixtral, Llama-2 and Falcon), the so-called judges, and analyze the impact on their performance of several factors, including confidence and authority of the source.
They found that susceptibility to influence is generally higher when the model is less certain about the correct answers (= when the model generally has a lower dataset accuracy) and when the explanations about the possible answers come from a source with a higher level of expertise (this is simulated by manipulating the root prompt of the LLMs, the test is carried out at six different levels of expertise of the "persona" of the model).

**Questions To Authors:**

TYPOS AND MINOR ISSUES:
- Section 3: "We do note... partially suppress the influence provided": this sentence is not very clear. Please, reformulate it.
- Section 3: aanchoring --> anchoring
- Section 3: This result, validates reported sycophantic behavior --> This result validates reported sycophantic behavior (but maybe in that context better "confirms the reported sycophantic behavior"?)
- Section 3.1: resolve around prompting --> revolve around prompting
- Section 3.1: byl the advocate --> by the advocate
- Section 4: in princliple -->  in principle

**Reasons To Accept:**

+ clear and exhaustive analysis of the impact of influence on LLM prompts, taking into account factors such as model confidence and source authority, and investigation of the possible mitigation strategies
+ narrative is generally easy to follow and experiments are well structured

**Reasons To Reject:**

- a possible issue is that the results all refer to a scenario in which the Llama-2 model is used as the advocate, and we cannot be sure about the generalizability of the findings (how would that change if a less powerful model was used? How about a more powerful model like GPT-4)? Also, in page 4 the authors should explain why they specifically chose to use Llama-2 as the advocate.

---

> ### Author Rebuttal · Authors · 2024-05-30
>
> We thank the reviewer for their in-depth review and the thoughtful comments and suggestions. We are pleased to hear that they found the narrative easy to follow and the analysis exhaustive and clear. In the following, we address the points raised.
>
> ---
> **Different Models as Advocate**
>
> That is indeed an interesting question. We point to our answer to reviewer YHyH (Section *More models and influence matrix*) for more results when using different models as advocates and judges. Briefly, the main takeaways are the following:
>
> 1) *Influence of more powerful advocates*: More powerful advocates are expected to generate more convincing explanations and thus lead to greater influence. This behavior is also observed in humans, who are more easily misled by more powerful models [1]. Our experiments confirm that this is the case, for LLM judges of different sizes. We anticipate that this trend will continue if even more powerful models, such as GPT-4, are used as advocators. In our research, we focused on open models, as many things remain unknown about closed commercial models.
>
> 2) *Susceptibility to influence with model size*: More powerful models are generally more susceptible to influence. This is likely due to the specifics of the training process and the training data used for alignment [2,3].
>
> In our experiments, we used *Llama-2-70b* as our advocate because it was the most powerful open-source model available that met our criteria for generating explanations for incorrect answers. More specifically, we found that although *Falcon* and *Mixtral* would sometimes refuse to create explanations for the wrong answer, this was not the case for the *Llama-2* model. We have provided a brief discussion in Appendix A, Section *Generated Explanations* on this topic. We made these details clearer in the revised manuscript.
>
> ---
> **Minor Issues**
>
> Thank you for the pointers and the suggestions. We updated the text to reflect them.
>
> ---
> We hope that in light of the above clarifications, we have adequately addressed your questions and concerns. In case you might have further doubts or comments, we remain at your disposal and welcome further discussion.
>
> ---
>
> [1] Michael, Julian, et al. "Debate helps supervise unreliable experts."
>
> [2] Perez, Ethan, et al. "Discovering language model behaviors with model-written evaluations."
>
> [3] Sharma, Mrinank, et al. "Towards understanding sycophancy in language models."

---

> > ### Comment · Reviewer_LLEW · 2024-06-04
> >
> > Thanks for your response! My concerns have been addressed, so I will keep the positive score.

---

### Official Review · Reviewer_9RHV · 2024-05-12

**Rating:** 5
**Confidence:** 4
**Ethics Flag:** 1

**Summary:**

The paper presents a study on the susceptibility of LLMs to the influence from additional prompts and information sources. The paper focuses largely on analyzing three LLMs (Llama2, Mixtral, and Falcon) and how adding other model generated answers to the prompt can influence their own answers. Nevertheless, there is no specific method proposed based on the analysis to mitigate the issue that authors point out. The findings on LLMs’ sensitivity to augmented inputs are interesting, but to my current understanding not as surprising.

The paper presents various figures to help illustrate its analysis and the writing is largely clear. Small typos need to be fixed, and the authors may want to make their summary or takeaways of each paragraph clearer.

**Questions To Authors:**

- To mitigate the issue of LLMs relying more on an answer/explanation in prompts that may even hurts its performance, have the authors tried fine-tuning the LLM on a training set of, say, CSQA with the augmented prompts? I would tend to believe that a small-scale fine-tuning (e.g. LoRA on 7B models) would largely solve this specific issue.
- It seems that LLama2 is used as the advocate. What would be the reason to make such specific choice? Would there be difference in terms of persuasiveness for different LLMs that serve as advocates?

**Reasons To Accept:**

- The paper presents an overall interesting analysis on LLMs’ susceptibility to prompts.
- The paper presents various analysis and visualizations to help illustrate the findings, which are quite helpful.

**Reasons To Reject:**

- It does not seem surprising that LLMs are sensitive to prompts, and a prompt stating that a certain option is the correct answer would almost surely have an impact on LLMs. Since they are autoregressive models and the next token generation depends on the previous tokens $p(x_{t+1}|x_1,\ldots, x_t)$, changing the context/prompts will cause differences in the model generations, without explicitly fine-tuning the model to disregard irrelevant or harmful contexts.
- Following my previous point, there have also been works that suggest LLMs tend to change their answers with external feedbacks (e.g. when the input states that one option is probably the correct answer), as the authors also listed some in the related works. Moreover, LLMs can generate persuasive adversarial prompts, causing other LLMs to produce incorrect or even unsafe answers (jailbreaking) [1]. Researchers have been studying the attack and defense for LLMs based on the fact that LLMs are highly susceptible to prompts without specific strategy/fine-tuning to make it more robust. Meanwhile, through the presentation of this work, I could not find a dedicated paragraph discussing the major difference in findings between this work and the previous works that discovered similar things.
- There are assumptions made in the paper that I feel is not well-supported (e.g. for PIQA, SIQA and CSQA, authors expect the model to have a strong bias and be strongly grounded?)
- Some part of the experiment setting is not well-explained. It is not clear that whether the authors used model-generated answer (and explanation) to augment the prompt, or pre-select an answer randomly and let model generate explanations to augment the prompt. (i.e. using model answers or random answers?)
- The last statement about having multiple advocates would improve the accuracy is reasonable. Meanwhile, it is highly similar to the concept of consistency and there are more works to leverage several LLM answers/reasonings for a more robust final answer.
- In terms of writing, I feel that it would be more helpful to have the takeaways summarized and highlighted in each section/paragraph that presents different findings.

[1] How Johnny Can Persuade LLMs to Jailbreak Them: Rethinking Persuasion to Challenge AI Safety by Humanizing LLMs

---

> ### Author Rebuttal · Authors · 2024-05-30
>
> We thank the reviewer for their in-depth review and their thoughtful suggestions.
>
> ---
> **Clarifications**
>
> We improved the clarity of the text.
> 1) *Strong bias*: For some datasets, unbiased performance is high, implying that the model is confident about many questions within [1] (see Fig. 2). Influence can then intuitively be expected to have a lesser effect.
> 2) *Model generated answers*: As stated in Section 2, for each question, we let the advocate generate explanations for each of the possible answers $\mathcal{Y}_i$.
>
> ---
> **Finetuning and takeaway message**
>
> We finetune using LoRA a *Llama-2-7b* model to ignore explanations based on conversations from the CSQA train set. We attach results [here](https://ibb.co/yfQPPPH), showing the influence of explanations belonging to wrong answers throughout training. Models finetuned to ignore explanations, successfully do so.
>
> Finetuning may be one of the possible ways to mitigate influence. However, finetuning can have additional effects, in terms of capabilities [4] (e.g. reduced instruction following capabilities) or safety [5]. A full study of such effects is out of the scope of our work, which aims to highlight the issue of excessive influence in current LLM judges. In general, further progress in reasoning is necessary. Whether this can be incorporated in the pretraining or alignment phase and how, are open problems that we hope our work will inspire. In short, this is also the main takeaway message of our study.
>
> ---
> **Consistency**
>
> We agree that there are connections between allowing multiple advocates and the concept of consistency [2,3], which we made more clear in the text. In our case, we are directly sampling probabilities for different answers (Eq. (3)), so we do not have any errors caused by sampling in the judge. We also do not allow the judge to perform complex reasoning. In our study, we are interested in understanding the anchoring bias that prompts can have in the answers selected by LLM judges, even when the reasoning provided is wrong (adversarial).
>
> ---
> **Different advocates**
>
> See our answer to reviewer YHyH (Section *More models and influence matrix*) for additional results.
>
> ---
> We hope that the above addresses the reviewer’s questions. We welcome any further discussion.
>
> ---
> [1] Mrinank, et al.
>
> [2] Xuezhi, et al. "Self-consistency improves ..."
>
> [3] Yanai, et al. "Measuring and improving ..."
>
> [4] Adam, et al. "Simple and scalable ..."
>
> [5] Xiangyu, et al. "Fine-tuning aligned ..."

---

> > ### Comment · Reviewer_9RHV · 2024-06-05
> >
> > Thank the authors for their rebuttal and the clarifications regarding certain points that I raised.
> >
> > Regarding different advocates, the authors provided additional experiments on Llama-2 models of different sizes as advocators or judges. I appreciate the additional results, but my question was more focused on the different model architectures (Llama/Mistral/Falcon/ect.) instead of model sizes.
> >
> > My concerns on the significance/impact of this work remain, where 1) I still consider the findings to be somewhat less surprising as LLMs are autoregressive models and will be affected by prompts and 2) the issue can be mitigated in many ways but not discussed or explored in this paper. If I understand correctly, the scope of this paper is to bring attention to existing issues of LLMs.  The analysis is indeed interesting but the finding itself does not seem surprising or difficult to deal with.
> >
> > I increased my rating accordingly with the efforts that authors made during rebuttal.

---

> > > ### Author Response · Authors · 2024-06-06
> > >
> > > Thank you very much for your continued engagement in the discussion and for allowing us to clarify our position further.
> > >
> > > ---
> > > **Other models as advocates**
> > >
> > > We agree that it would be interesting to explore other models as advocates. In our original experiments, we focused on *Llama-2* because we found it to be the most “compliant” with our request to generate explanations for incorrect answers. Specifically, while models like *Falcon* and *Mixtral* occasionally refused to create explanations for incorrect answers, this was not the case for the *Llama-2* model. We point to a brief discussion on this topic in Appendix A, Section “Generated Explanations”. For further details, we also refer to our response to reviewer LLEW.
> > >
> > > ---
> > > **LLMs are affected by prompts**
> > >
> > > We agree with the reviewer that the fact that LLMs are influenced by prompts is not by itself surprising. After all, these models are notorious for their in-context learning [1]. Our work aims to highlight the degree of this influence when opinions are presented in the prompts. We found that models are overly relying on these opinions presented, leading to substantial performance deterioration. This is particularly significant as these models are increasingly being used as judges in various evaluation frameworks [2, 3].
> > >
> > > ---
> > > **Mitigation techniques**
> > >
> > > Our experiments reveal that while LLMs are indeed influenced by the content of their prompts, standard prompting techniques do not effectively mitigate this influence. Specifically, neither chain-of-thought nor few-shot prompting resulted in a significant reduction in this effect. This is a critical highlight of our paper. We believe that exploring alternative approaches, such as fine-tuning, may offer more promising results in reducing this influence. By bringing this phenomenon into the spotlight, we hope to encourage further research on effective mitigation strategies in this area.
> > >
> > > ---
> > > Thank you again for your valuable feedback. We hope these clarifications address your concerns.
> > >
> > > ---
> > > References
> > >
> > > [1] Dong, Qingxiu, et al. "A survey on in-context learning." arXiv preprint arXiv:2301.00234 (2022).
> > >
> > > [2] Chang, Yupeng, et al. "A survey on evaluation of large language models." ACM Transactions on Intelligent Systems and Technology 15.3 (2024): 1-45.
> > >
> > > [3] Li, Xuechen, et al. "Alpacaeval: An automatic evaluator of instruction-following models." (2023).

---

### Official Review · Reviewer_YHyH · 2024-05-14

**Rating:** 7
**Confidence:** 3
**Ethics Flag:** 1

**Summary:**

This paper studies the sensitivity of LLMs’ responses to the injected context in the prompt. It examines how the output response of an LLM will be swayed by the prediction and explanations provided by an external source. It shows that models are generally easily swayed by the explanations, regardless of their quality. And when the input is presented as being authoritative or confident, the model is more easily swayed.

**Questions To Authors:**

See the above section.

**Reasons To Accept:**

This paper studies an important problem in LLMs: the influence of the external knowledge sources (or context) on the LLM responses. Such a problem is particularly important in retrieval augmented generation settings, where there is potential conflict between the external context and the internal knowledge. Examining the model’s robustness against authoritative advocacy and possibly erroneous retrieved context is important to build an LLM system that is trustworthy.

This paper carries out a comprehensive study of the problem and shows that LLMs are easily swayed by the injected context from an advocating model, especially when explanations are present there. In particular, the paper examines the influence of context injection from different aspects, such as the confidence of the advocacy and the confidence of the judge model. In addition, it further shows that different prompting methods have limited effect in mitigating the issue and calls for further measures for solving the problem.

**Reasons To Reject:**

This paper mainly examines the influence of injected context on QA tasks. Advocacy would more influence the potential factuality. It would be interesting to see how such advocating context would influence reasoning capabilities of LLMs. In particular, how would it influence the factuality and correctness of the step-by-step reasoning processes. How resilient the reasoning process is against such injection?

It also shows that critical reasoning ability is insufficient to address the problem. It is also interesting to see whether a model with stronger general logic reasoning capability would be more robust against such advocacy? The three models used in the paper would be close in their capabilities. It would be interesting to see how the advocating context would influence a wider spectrum of models from weak reasoning capabilities (7B or even smaller models) to much stronger reasoning capabilities (e.g., Llama3-70B). Also, it would be interesting to see whether a stronger model would have much stronger influence on the judge model and vice versa. Overall, a “influence matrix” that characterizes different levels of advocating model vs different levels of judge model would help draw a more complete picture of the problem. Currently, it is mainly through prompting the model to be different levels. However, using models with different levels of actual capabilities would be more interesting.

---

> ### Author Rebuttal · Authors · 2024-05-30
>
> We thank the reviewer for their in-depth review. Below we address the questions raised.
>
> ---
> **Reasoning capabilities**
>
> Two key findings of our study:
> 1) LLMs are highly susceptible to influence even without explanations.
> 2) When explanations are provided, LLM judges can often identify flawed reasoning, especially for datasets where the model exhibits higher unbiased performance.
>
> We conducted additional experiments on the *gsm8k* dataset, which requires step-by-step reasoning. We adapted the dataset to our QA format and randomly sampled $3$ numbers ($1-1000$) to serve as alternative answers. We then asked a *Llama-2-70b* advocate to generate explanations for these answers. Given that alternative answers were randomly generated, the explanations often contained obviously wrong reasoning. We use a *Llama-2-70b* as a judge.
>
> With explanations advocating for the wrong answer, the influence score was $0.16$, compared to $0.98$ when no explanations are given. This significant difference (see also Fig. 3 for differences in the others datasets), indicates that models can identify clearly wrong reasoning.
>
> ---
> **More models and influence matrix**
>
> This is a great suggestion. We anticipate that better explanations, by more capable advocates, will lead to higher influence. This is also observed in human behavior, as more capable models are better at misleading human judges [1]. Additionally, we expect larger models to be more easily influenced due to their higher levels of sycophantic behavior [2,3], likely a direct cause of the training processes and data used during alignment. This may change as new alignment techniques are developed. We conducted the following experiment; we use different *Llama* models as advocators or judges and compute the influence averaged across the datasets used in our study:
>
> |Advocate\Judge|*Llama-2-7b*|*Llama-2-13b*|*Llama-2-70b*|
> |----------------------|------------------|-------------------|-------------------|
> |*Llama-2-7b*|0.668|0.686|0.918|
> |*Llama-2-13b*|0.682|0.697|0.928|
> |*Llama-2-70b*|0.712|0.721|0.927|
>
> Results adhere to our hypotheses. We believe our framework provides a valuable starting point for a more rigorous evaluation of LLM susceptibility to influence.
>
> ---
> We thank the reviewer for their insightful suggestions, which have enriched our evaluation framework. We hope the above responses address the reviewer's questions and welcome further discussion.
>
> ---
> [1] Julian, et al.
>
> [2] Ethan, et al.
>
> [3] Mrinank, et al.

---

### Decision · Program_Chairs · 2024-07-10

**Decision:**

Accept

**Comment:**

This paper investigates the susceptibility of LLMs to influence from prompts, focusing on how additional context (especially explanations and confidence) from another model affects LLM responses in QA tasks.

Strengths:
1. The paper studies a timely topic, and the impact of external knowledge sources on responses is very meaningful for downstream system building (Reviewer YHyH, LLEW).
2. The paper conducted a comprehensive analysis of the impact of context injection from various aspects (Reviewer LLEW).

Weaknesses:
1. Experiment - The study mainly examines QA tasks and therefore may have limited scope (Reviewer YHyH). Further, some experiments seem to have made too strong assumptions on the dataset and the model (Reviewer 9RHV).
2. Experiment - The study mostly relied on Llama-2 as the advocate, which made the reviewers question the generalizability to other model sizes (Reviewer YHyH) as well as other model architectures (Reviewer 9RHV).
3. Contribution - The paper mostly focused on raising the issue, but did not propose a method to mitigate the influence of prompts (Reviewer 9RHV).
4. Contribution - The prompt sensitivity is not by itself surprising and the contribution may be considered less substantial (Reviewer 9RHV)

In their rebuttal, the authors (1) added new experiments using different sizes of Llama-2 as advocate and judge for extending the generalizability consideration, (2) provided hypotheses on further generalization to more capable models like GPT-4, (3) explained why Llama was picked as the main model architecture. Most importantly, they distinguished their work from others on prompt sensitivity: "Our work aims to highlight the degree of this influence when opinions are presented in the prompts. We found that models are overly relying on these opinions presented, leading to substantial performance deterioration. This is particularly significant as these models are increasingly being used as judges in various evaluation frameworks"

With this distinction, I believe the paper makes sufficient enough contribution to the field, though I do agree this could have been made more clear even e.g., in the paper title.